# Estimating the Ripeness of Hass Avocado Fruit Using Deep Learning with Hyperspectral Imaging

Yazad Jamshed Davur [1] , Wiebke Kämper [2] , Kourosh Khoshelham [3] , Stephen J. Trueman [2] and Shahla Hosseini Bai [2],*

1   School of Computing and Information Systems, The University of Melbourne, Parkville, VIC 3010, Australia; yazadjd@yahoo.com
2   Centre for Planetary Health and Food Security, School of Environment and Science, Griffith University, Nathan, QLD 4111, Australia; w.kaemper@griffith.edu.au (W.K.); s.trueman@griffith.edu.au (S.J.T.)
3   Department of Infrastructure Engineering, The University of Melbourne, Parkville, VIC 3010, Australia; k.khoshelham@unimelb.edu.au
*   Correspondence: s.hosseini-bai@griffith.edu.au

**Abstract:** Rapid ripeness assessment of fruit after harvest is important to reduce post-harvest losses by sorting fruit according to the duration until they become ready to eat. However, there has been little research on non-destructive estimation of the ripeness and ripening speed of avocado fruit. Unlike previous methods, which classify the ripeness of fruit into a few categories (e.g., unripe and ripe) or indirectly estimate ripeness from its firmness, we developed a method using hyperspectral imaging coupled with deep learning regression to directly estimate the duration until ripeness of Hass avocado fruit. A set of 44,096 sub-images of 551 Hass avocado fruit images was used to train, validate, and test a convolutional neural network (CNN) to predict the number of days until ripeness. Training, validation, and test samples were generated as sub-images of Hass fruit images and were used to train a spectral–spatial residual network to estimate the duration to ripeness. We achieved predictions of duration to ripeness with an average error of 1.17 days per fruit on the test set. A series of experiments demonstrated that our deep learning regression approach outperformed classification approaches that rely on dimensionality reduction techniques such as principal component analysis. Our results show the potential for combining hyperspectral imaging with deep learning to estimate the ripeness stage of fruit, which could help to fine-tune avocado fruit sorting and processing.

**Keywords:** avocado; deep learning; Hass; hyperspectral imaging (HSI); post-harvest; ripening

## 1. Introduction

Inadequate supply of human food and nutrition has long been recognized as a major problem in many parts of the world [1]. Food and nutrition are fundamental for physical and cognitive development and a well-functioning immune system [2]. Increasing the production of nutrient-rich foods is important, but it is also essential to improve the food supply chain by reducing food loss and minimizing food waste. Currently, one-third of food produced for human consumption is lost or wasted globally [3]. Post-harvest food loss represents both a nutrition loss and an economic loss [4]. Two of the main reasons for post-harvest food loss are over-ripening and rancidity, which often cannot be detected visually [3]. Visual estimation of ripeness in some fruits is also inaccurate and inefficient [5]. Rapid and accurate quality and ripeness assessment methods need to be developed to reduce the loss of food.

Hyperspectral imaging (HSI) is emerging rapidly as a novel tool for the non-invasive classification of food quality, utilizing spatial and spectral information of the sample [6,7]. Hyperspectral data are represented by a three-dimensional cube with two spatial dimensions and one spectral dimension [5]. Importantly, HSI captures reflectance values across

the visible (VIS) and the near infrared (NIR) wavelengths of the electromagnetic spectrum [8]. The spectral data contained within a spatial pixel of a HSI cube provide a unique spectral signature of the sample which, in turn, provides information about its quality characteristics. For this reason, HSI is utilized frequently for food quality inspection. In recent years, HSI-based analyses have become a popular non-destructive method to estimate the quality of agricultural products including fruit, meat, and vegetables [9–16]. HSI analysis has been used to determine the firmness of strawberries, peaches, and many other fruit [12,13]. VIS/NIR–HSI (400–1110 nm) has been used to predict beef color parameters and tenderness [14]. Contamination of citrus fruits with *Penicillium fungi* has been detected using machine learning combined with HSI [15]. HSI has also been used to estimate the degree of rancidity of nuts using deep neural networks [16]. However, HSI-based technology is underutilized for quality assessment of tropical fruit.

Among the many tropical fruit, avocado fruit have numerous health benefits [17]. They have high nutritional density and provide major antioxidants, fruit proteins, and fiber [18–21]. They can also help with human weight control and stroke prevention [17]. Avocado fruit reach full maturity on the tree but do not ripen on the tree [5]. Hence, mature avocado fruit are harvested from the tree canopy and are then ripened, partly or completely, prior to retail display. The time needed to obtain full ripeness varies, even among fruit collected from the same tree [22]. This variability can potentially lead to major fruit losses due to over-ripening. A post-harvest management method is required to determine the duration until ripeness so that fruit can be sorted into homogenous groups that ripen simultaneously. Homogenous groups would provide better control over the supply chain and help to guarantee fruit quality for consumers, thus reducing waste [5].

Using imaging technologies to predict internal fruit quality may be challenging for fruit with thick skin [23]. However, HSI has been applied successfully to predict dry matter concentration of Hass avocado fruit from images of the skin [23–25]. Dry matter concentration correlates strongly with the oil concentration and maturity of avocado fruit and is thus used to determine the harvest time [23]. Fatty acid composition and mineral nutrient concentrations can also be predicted from flesh and skin images of Hass fruit [26]. Therefore, there is potential to predict the internal quality of avocado flesh using skin images, and these might also be used to predict the duration to fruit ripeness.

Avocado fruit are harvested when the fruit are mature but ripening occurs after harvest when fruit are taken out of cool storage and placed on shelves after harvest. It is important to predict the duration to ripeness from mature avocado fruit or from fruit at subsequent ripening stages. The ripening stage of Hass avocado fruit has been predicted non-invasively through smartphone images and hyperspectral images [27–30]. However, these approaches either predict the ripeness indirectly by estimating the firmness of the fruit [31,32], or by classifying the fruit into a very limited number of ripeness categories of unripe, ripe, and over-ripe [5,29,33]. Avocado fruit ripening is highly variable and ripening time may vary between 6 and 15 days when fruit are placed on shelves after harvest. Therefore, classifying a fruit into an unripe category would not suggest how many days that the fruit needs to ripen. In our recent study, we were able to predict the ripening time of Hass and Shepard avocado fruit when images were captured only once from mature fruit after harvest [30]. Predicting the ripening time of mature fruit after harvest allows fruit classification on the farm before sending the fruit to retail stores. However, it is also important to be able to predict the ripening time at retail stores when the fruit are placed on display with an unknown duration from harvest. Therefore, we aimed to determine whether the duration that a Hass avocado fruit takes to ripen could be estimated directly through machine learning models that are applied to hyperspectral images of the fruit and, if so, with what accuracy. We developed a deep learning regression model that takes advantage of the abundance of information present within hyperspectral images to predict the time required to ripen avocado fruit. The research makes two novel contributions: (a) we performed regression over a deep neural network, which was different from previous approaches that are largely based on classification; and (b) we performed

experiments involving conventional classification techniques using PCA and compared the results with the regression approach. Through these experiments, we demonstrate that deep regression performs better than classification and should be considered the preferred approach for estimating the ripeness of Hass avocado fruit. This is in contrast to the common understanding of deep learning, as reported in the literature, where classification is considered a simpler problem than regression and deep learning classification methods are considered more successful than their regression counterparts. These findings will potentially help to increase the efficiency of post-harvest processing, improve the quality of avocado fruit at retail outlets, and reduce post-harvest losses in the agri-food industry.

## 2. Materials and Methods

### 2.1. Site Description

We collected Hass avocado fruit in dry weather from two irrigated orchards (25°13′32″ S 152°17′53″ E and 25°08′17″ S 152°22′40″ E) in Queensland, Australia, in April and June 2018. The soil in the orchards is red clay-loam. The sites receive average precipitation of 1004 mm annually. The average maximum daily temperatures varied between 22.3 °C and 31.4 °C, and the average minimum daily temperatures ranged between 11.0 °C and 22.4 °C, in 2018 (Bureau of Meteorology 2023).

### 2.2. Sample Collection and Preparation

Ten mature Hass avocado fruit were harvested from each of eight trees, providing eighty fruit in total. Each tree was divided into five sectors, with one fruit harvested from the inside and one fruit harvested from the outside of the canopy in each sector. The fruit were transferred to a refrigerated room at 4 °C after harvest. The recommended storage temperature for Hass fruit is approximately 4–6 °C [32,33]. Fruit were kept in the refrigerated room for 10 or 20 days, before being stored at approximately 21 °C to let them ripen. We confirmed ripeness by assessing skin firmness with a hand-held sclerometer (8 mm head; Lutron Electronic Model: FR-5120, Coopersburg, PA, USA). A fruit was considered ripe when the maximum force required to impress the sclerometer tip 1 mm deep was <15 N for the skin [32,33]. The fruit were ripe after 10.6 ± 1.0 days (mean ± SE) at room temperature. The number of days until ripeness was recorded for each individual avocado, allowing us to match each hyperspectral image with the duration until ripeness.

### 2.3. Imaging System

We acquired hyperspectral images using a laboratory-based 12-bit line scanner camera (Pika XC2, Bozeman, MT, USA) with a 2.3 cm focal-length lens. The camera had a spectral resolution of about 1.3 nm and produced 462 bands in the wavelength range, 388.9–1005.33 nm. There were four current-controlled wide-spectrum quartz–halogen lights for illumination. We placed each sample on a black tray on the translation stage and imaged each individual fruit every day until it was fully ripe and the skin color was dark purple rather than green (Figure 1). In total, 551 images were obtained. Some fruit became ripe after 6 days at 21 °C and so the number of images captured per day decreased from this day onward (Figure 2). The exposure time was adjusted to 19.4 ms. Image capture and data extraction were performed using Spectronon Pro software (Version 2.112; Resonon, Bozeman, MT, USA).

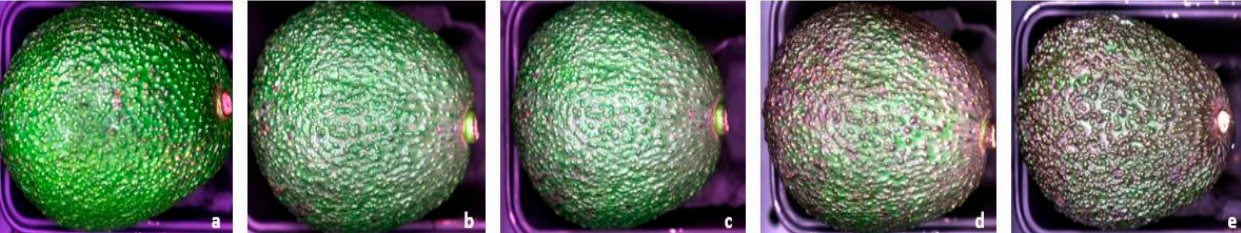

**Figure 1.** RGB (red, green, blue) images generated through hyperspectral imaging of a single Hass avocado fruit with (**a**) 14 days left to ripen; (**b**) 10 days left to ripen; (**c**) 6 days left to ripen; (**d**) 2 days left to ripen; (**e**) 0 days left to ripen (completely ripe).

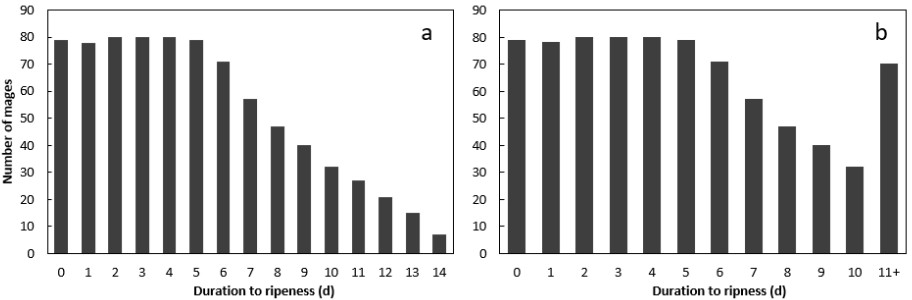

**Figure 2.** (**a**) Number of original hyperspectral images of Hass avocado fruit in each ripeness-stage category; (**b**) number of hyperspectral images of Hass avocado fruit in each category after combining the categories with 11–14 days to ripeness.

We set the pixel value to the mean corrected relative reflectance, which was calculated using Equation (1):

$$R = (R_0 - D)/(W - D) \tag{1}$$

where $R_0$ is the raw spectral reflectance, D is the reflectance of a pure dark image captured by the same camera with its lens covered, and W is the reflectance of a pure white Teflon sheet that reflected approximately 99% of incident light [11]. This adjustment was performed to correct for the spectral curve of the sample surface. The reflectance data were scaled up by 10,000 automatically by the software. We captured images from 80 different fruit daily, a total of 551 images, until each fruit was found, by using the sclerometer, to be completely ripe.

### 2.4. Generation of Training, Validation, and Test Samples

The acquired hyperspectral images were partitioned randomly into three sets: training, validation, and testing. The training set was used to train the deep learning model, the validation set was used to ensure the model did not overfit the training data, and the test set was used to evaluate model performance in estimating the number of days to ripeness. An implicit assumption was that the entire fruit was ripe after the same number of days and so any spatial variation was ignored.

The original size of the HSI images was very large (1600 × 1 × 462 pixels) and included part of the tray on which the samples were placed (Figure 1). Additionally, a 551-image sample size was considered small for training a deep neural network. Therefore, sub-images were extracted using Envi (Version 5.5.3) and Interactive Data Language (IDL) (Version 8.7.3) to eliminate the background and extract more training data from each HSI image. Envi is an industry-standard spectral image processing and analysis software package that is written in IDL. We defined small regions of interest (ROIs) around the center of the fruit image, ignoring the background, using Envi. These ROIs were then extracted as sub-images by executing a script in IDL.

The IDL script was programmed to generate 60 sub-images from each individual image and these were allocated randomly into 40 training samples, 8 validation samples, and

12 test samples (Table 1). Each of the extracted sub-images was unique and had no overlap with neighboring sub-images. We set the size of each sub-image to 50 pixels × 50 pixels. This led to a total of 29,392 training images, 5872 validation images, and 8832 test images, each of 50 × 50 × 462 pixels (Table 1). The label of each sub-image (i.e., number of days to ripen) was inherited from its original full HSI image.

**Table 1.** Number of sub-images of Hass avocado fruit in the training, validation, and test datasets.

|  | Training | Validation | Test |
|---|---|---|---|
| Number of sub-images generated for each sample | 40 | 8 | 12 |
| Total number of sub-images | 29,392 | 5872 | 8832 |

The number of samples in each category from 0 days to ripen to 14 days to ripen was unbalanced (Figure 2a). Therefore, all the sub-images in the categories, 11, 12, 13, and 14 days to ripen, were combined into a single category (Figure 2b) for the classification experiments to balance the dataset. In all classification experiments, the input labels were one-hot encoded, allowing the representation of categorical labels to be more representative of the actual categories.

### 2.5. Deep Learning Approach

Two major attributes of hyperspectral imaging need to be considered to obtain discriminative features: (1) ample spectral information [34] which, in our study, was extracted from 462 contiguous spectral bands; and (2) spatial features that originate from homogenous areas within the hyperspectral image [35]. To take advantage of abundant spectral bands, traditional pixelwise HSI classification models mainly concentrate on feature engineering and classifier training [36]. The main objectives of feature engineering are to reduce the high dimensionality of HSI pixels and extract the most discriminative features or bands. The classifiers are then trained using these extracted features [36,37]. Although these traditional classification frameworks are used frequently, they have some limitations. Firstly, the feature engineering step might not generalize well to all categories. Secondly, the default one-layer, non-linear transformation applied before classification has limited representation capacity to fully utilize the abundant spectral and spatial features [36].

### 2.6. Network Architecture

We experimented with two different types of networks: a regression-based network and a classification-based network. We adopted a spectral–spatial residual network (SSRN), developed originally for satellite images, to perform the backbone feature extraction for each network [36]. The network itself included consecutive learning blocks that took the major characteristics of hyperspectral images into consideration. The designed spectral and spatial residual blocks extract discriminative spectral–spatial features from HSI cubes and can be regarded as an extension of convolutional layers in convolutional neural networks (CNNs). The shortcut layers between every other convolutional layer allow the SSRN to learn from the original HSI image.

A major challenge of CNN models used in our previous research is the large number of learnable parameters in the convolutional layers, which requires a large set of training samples to adequately train the network [30]. In practice, training data are scarce, due to the cost and labor-intensiveness of manually labeling hyperspectral images of avocado fruit. In the current work, we used the SSRN because the SSRN has the ability to allow the network to learn from both the spectral features, which represent the reflectance properties of the fruit across the wavelength band, and the spatial features, which represent the variation in spectral reflectance across the fruit surface. In addition, the SSRN implements residual/shortcut links to alleviate the decreasing-accuracy caused by an increasing number of convolutional layers [38], and applies batch normalization (BN) to prevent overfitting unbalanced training data. Thus, SSRNs have frequently achieved state-of-the-art classifica-

tion accuracy on HSI datasets using limited training data, making it an attractive network for adoption.

The adopted SSRN in our study had a spectral feature learning module that included two convolutional layers and two spectral residual blocks (Figure 3). This module was followed by the spatial feature learning module that included a 3-D convolutional layer and had two spatial residual blocks. Following the two feature learning modules, an average pooling layer transformed the feature volume to a feature vector and then a fully connected (FC) layer adapted to the residual network of the dataset according to the number of categories or output units. A dropout layer was also included after the average pooling layer for appropriate regularization (Figure 3).

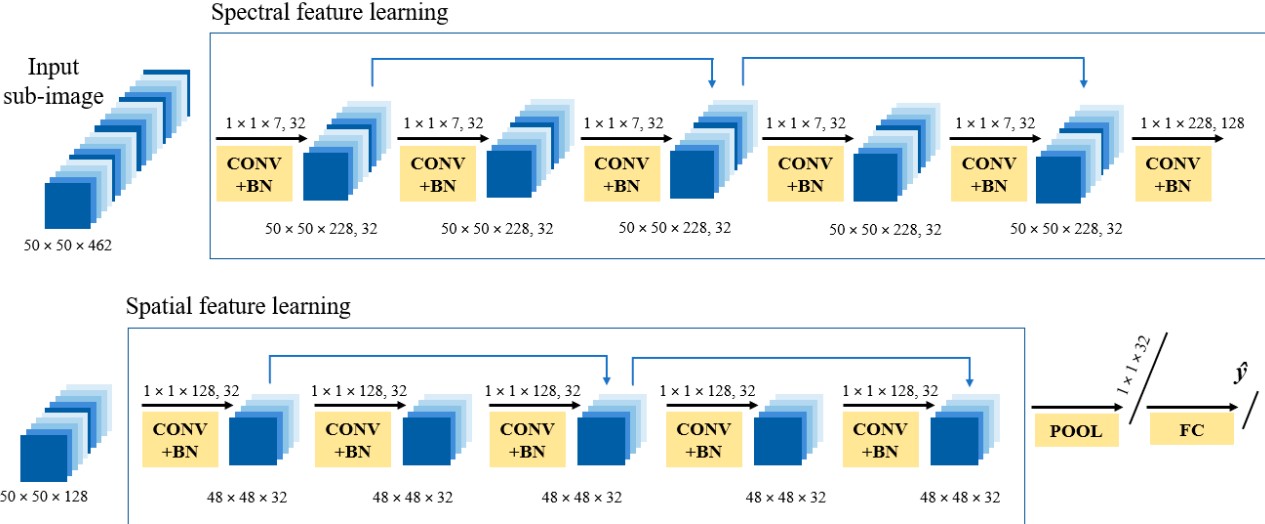

**Figure 3.** Spectral–spatial residual network architecture with 50 × 50 × 462 input volume (Z). CONV + BN denotes a convolutional layer with batch normalization, POOL denotes an average pooling layer, and FC denotes a fully connected layer that outputs a single continuous value $\hat{y}$.

### 2.7. Regression

We developed a regression model using the underlying SSRN architecture. Input labels were fed into the network in the form of integer values of the number of days to ripeness. Additionally, the number of output units in the fully connected layer was a single unit with a linear activation function, resulting in a single continuous value.

The regression model was trained by minimizing the loss function, defined as the mean squared error (MSE) loss, as shown in Equation (2):

$$\text{MSE} = \frac{1}{n} \sum_{i=1}^{n} \left( Y_i - \hat{Y}_i \right)^2 \tag{2}$$

where n is the number of data points, $Y_i$ are the observed/actual ground truth values, and $\hat{Y}_i$ are the predicted values.

We were able to use a regression model because the labels used from 0–11 "days to ripeness" were not distinct categories, but had an ordinal relationship between them, which is not taken into consideration by a cross-entropy-based softmax classifier.

### 2.8. Classification

The softmax classifier has a distinctive advantage when managing N-dimensional vectors and has been widely used in deep learning with the rapid development of computer vision [39]. In the model that we adopted here, the feature vectors of all samples were extracted by training a softmax model for the classification-based experiments.

The softmax score function gives a specific probability of mapping based on the final score. The sum of the probabilities of all categories is 1. The function form is shown in Equation (3) [40]:

$$\mathbf{f_j}(\mathbf{z}) = \frac{e^{z_j}}{\sum_k e^{z_k}} \tag{3}$$

where the $z_j$ values are the elements of the input vector and the denominator is the normalization term to ensure that all the output values of the function will result to 1, constituting a valid probability distribution.

The softmax loss function mainly uses the form of cross-entropy loss, which can be seen as the entropy of two probabilities, as shown in Equation (4) [40]:

$$H(p,q) = -\sum_x p(x) \log(x) \tag{4}$$

where $p$ represents the probability of true classification and $q$ represents the probability of the predicted classification. The loss function measures the size of the error between the true classification result and the predicted classification result.

### 2.9. Implementation Details

The raw input contained rich and redundant spectral information. The input layer also had high dimensionality that needed to be reduced. Hence, we set the convolutional layers in the spectral feature learning module to contain 32 kernels with a size of $1 \times 1 \times 7$ to allow the extraction of low-level and deep spectral features of the image for consecutive layers. The last convolutional layer in this module, however, had 128 kernels, with a size of $1 \times 1 \times 128$ to keep discriminative spectral features before it was sent as input to the spatial learning module.

The first convolutional layer used in the spatial learning module contained 32 kernels each with size $1 \times 1 \times 128$, which extracted low-level spatial features and reduced the input size of the feature cubes. We set a kernel size of $1 \times 1 \times 32$ for the following convolutional layers within the residual blocks, keeping the size of the feature cubes unaffected.

We used the ReLU activation function, defined below in Equation (5), in the SSRN due to its benefits, as highlighted previously [41]:

$$\sigma\left(w^T x\right) = \mathbf{max}\{\mathbf{0}, w^T x\} \tag{5}$$

where $x$ is the input and $w$ is the weight parameter learned using back-propagation [42].

The hyperparameters of the optimal SSRN used for learning are shown below (Table 2). The weights were initialized by drawing samples from a truncated normal distribution (HeNormal) centered on 0 with a standard deviation as in Equation (6) [43]:

$$\text{stddev} = \text{sqrt}\left(\frac{2}{\text{fan}_{in}}\right) \tag{6}$$

where $\text{fan}_{in}$ is the number of input units in the weight tensor.

**Table 2.** Hyperparameters for the optimal SSRN.

| Parameter | Value/Type |
|---|---|
| Weight initializer | "HeNormal" |
| Optimizer | SGD |
| Learning rate | 0.01 |
| Batch size | 32 |
| Number of kernels | 32 |
| Spatial input size | $1 \times 1$ |

We developed our input pipeline and performed all experiments and evaluations using the Keras deep learning library in Python. Keras allowed us to focus on the main

concepts of deep learning while managing the fine details of tensors, their shapes, and their mathematical elements [44].

Due to the high number of pixels in each training image ($50 \times 50 \times 462$), batchwise model training was conducted using the University of Melbourne's SPARTAN High Performance Computing (HPC) facility [45]. This allowed for distributed synchronous training on four separate P100 Nvidia Graphics Processing Units (GPUs) simultaneously on the HPC.

### 2.10. Evaluation Metrics

The evaluation metric used for the regression model was the root mean squared error (RMSE), which is the square root of the MSE, as given in Equation (2), calculated over all test samples. MSE is a measure of the degree to which the regression line fits the data. The error is the mean deviation of the prediction from the true value [46]. A mean RMSE of 2, for example, indicates a deviation of 2 days to ripeness on average for the batch of samples under study.

The evaluation metric used for the classification experiments was categorical accuracy, given by Equation (7). It is defined as the mean classification accuracy across all predictions.

$$\text{Accuracy} = \frac{c_0 + c_1 + \ldots + c_{11}}{n} \qquad (7)$$

where $c_0$, $c_1$, ..., $c_{11}$ are the number of correctly classified samples of avocado fruit in categories 0, 1, ..., 11 "days to ripen", and $n$ is the total number of samples tested.

### 2.11. Ablation Study

We compared the backbone SSRN with that of a DenseNet neural network. DenseNet is used extensively for image classification problems due to numerous advantages. For example, DenseNet maximizes the flow of information between layers since all layers are connected directly with each other in the network [47]. It also encourages feature reuse and alleviates the vanishing gradient problem. We further compared the performance of the classifier and the regressor on each of the above network architectures and assessed the effect of applying dimensionality reduction through PCA (10 bands). Each of these experiments was performed without any change in other framework settings.

## 3. Results

### 3.1. Regression

Training loss was minimized and converged within 200 epochs in the regression model (Figure 4). The root mean squared error (RMSE) on all samples of the test dataset (estimation) was 1.32, indicating an overall mean deviation of 1.32 days to ripeness from the actual value. We also computed the RMSE value for each whole fruit image from the corresponding sub-images and calculated the mean RMSE per fruit sample to be 1.17 days (~28 h). The avocado fruit with maximum misprediction had an RMSE value of 3.63 days to ripeness and the fruit with the closest prediction had an RMSE value of 0.17 days to ripeness (~4 h) (Table 3). These findings demonstrated that the regression-based model could estimate the number of days to ripeness with an uncertainty of, on average, 1.17 days per fruit.

**Table 3.** RMSE values after 200 epochs for the regression model for predicting ripeness of Hass avocado fruit.

| Evaluation Set | RMSE (Days) |
|---|---|
| RMSE over all sub-images | 1.32 |
| Average RMSE per fruit sample | 1.17 |
| Minimum RMSE per fruit sample | 0.17 |
| Maximum RMSE per fruit sample | 3.63 |

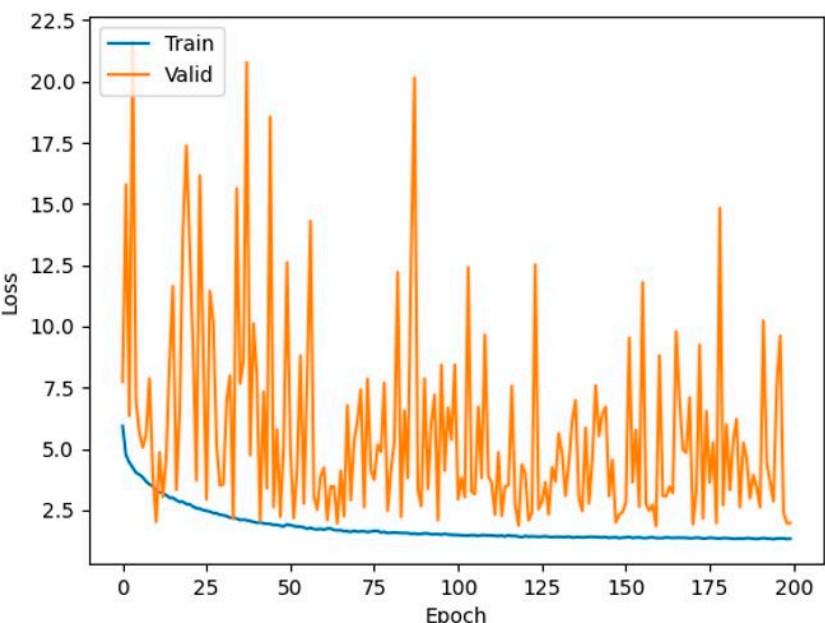

**Figure 4.** Loss curves of training (blue) and validation (orange) datasets of the mean squared error of the regression model for predicting the duration to ripeness of Hass avocado fruit.

*3.2. Classification*

The classification network was trained for 1400 epochs (Figure 5). The overall mean prediction accuracy in the classification model on the test dataset across the 12 categories from 0–11 days to ripeness was 51.43% (Table 4). The minimum accuracy per individual fruit was 0%, indicating that there were few fruit where all sub-images of the test dataset were completely misclassified (Table 4). However, the maximum prediction accuracy per fruit was 100%, indicating that there were also fruit where all test sub-images had been correctly classified (Table 4). The confusion matrix for the samples in the test dataset indicated that the first few categories from 0–5 days to ripeness were predicted more accurately than the categories from 6–11 days to ripeness (Table 5). However, the prediction accuracy was still much higher than that of a random estimator (51.43% vs. ~8.34%). To better understand the performance of the classification model, we visualized the output feature maps of the model, i.e., the output from the model that went into the classifier (Figure 6). We used a dimensionality reduction technique called t-distributed stochastic neighboring entities (t-SNE) to visualize this in two dimensions [48].

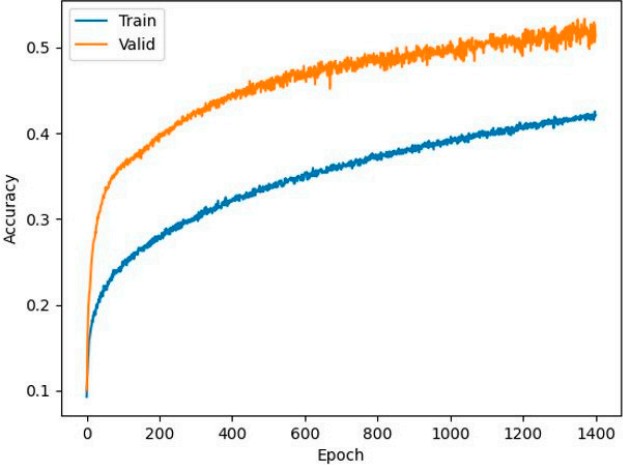

**Figure 5.** Prediction accuracy curves of the training (blue curve) and validation (orange curve) datasets for the classification model for predicting the duration to ripeness of Hass avocado fruit.

**Table 4.** Accuracy values after 1400 epochs for the classification model for predicting the duration to ripeness of Hass avocado fruit.

| Evaluation Set | Accuracy |
|---|---|
| Accuracy over all test sub-images | 51.43% |
| Average accuracy per fruit sample | 51.43% |
| Minimum accuracy per fruit sample | 0.00% |
| Maximum accuracy per fruit sample | 100.00% |

**Table 5.** Confusion matrix of duration to ripeness for samples of the test dataset predicted using the classification model. The numbers in green indicate the number of correct predictions for that category while the other values are false results.

| Duration to Ripeness (d) | 0 | 1 | 2 | 3 | 4 | 5 | 6 | 7 | 8 | 9 | 10 | 11 |
|---|---|---|---|---|---|---|---|---|---|---|---|---|
| 0 | 729 | 67 | 0 | 0 | 9 | 0 | 0 | 0 | 0 | 0 | 0 | 0 |
| 1 | 143 | 594 | 0 | 1 | 112 | 2 | 0 | 0 | 0 | 0 | 0 | 0 |
| 2 | 0 | 0 | 54 | 217 | 0 | 1 | 0 | 4 | 24 | 8 | 38 | 38 |
| 3 | 0 | 0 | 10 | 750 | 0 | 0 | 0 | 0 | 27 | 6 | 40 | 19 |
| 4 | 3 | 95 | 0 | 0 | 600 | 143 | 20 | 3 | 0 | 0 | 0 | 0 |
| 5 | 0 | 8 | 0 | 2 | 149 | 527 | 190 | 22 | 1 | 0 | 0 | 1 |
| 6 | 0 | 1 | 0 | 2 | 4 | 216 | 415 | 191 | 53 | 4 | 1 | 0 |
| 7 | 0 | 1 | 0 | 5 | 1 | 43 | 200 | 475 | 140 | 10 | 8 | 5 |
| 8 | 0 | 0 | 10 | 20 | 1 | 10 | 85 | 286 | 291 | 33 | 11 | 9 |
| 9 | 0 | 0 | 7 | 26 | 0 | 2 | 24 | 151 | 247 | 55 | 56 | 44 |
| 10 | 0 | 0 | 27 | 93 | 1 | 0 | 9 | 54 | 202 | 31 | 74 | 61 |
| 11 | 0 | 0 | 29 | 122 | 0 | 0 | 1 | 10 | 131 | 35 | 54 | 98 |

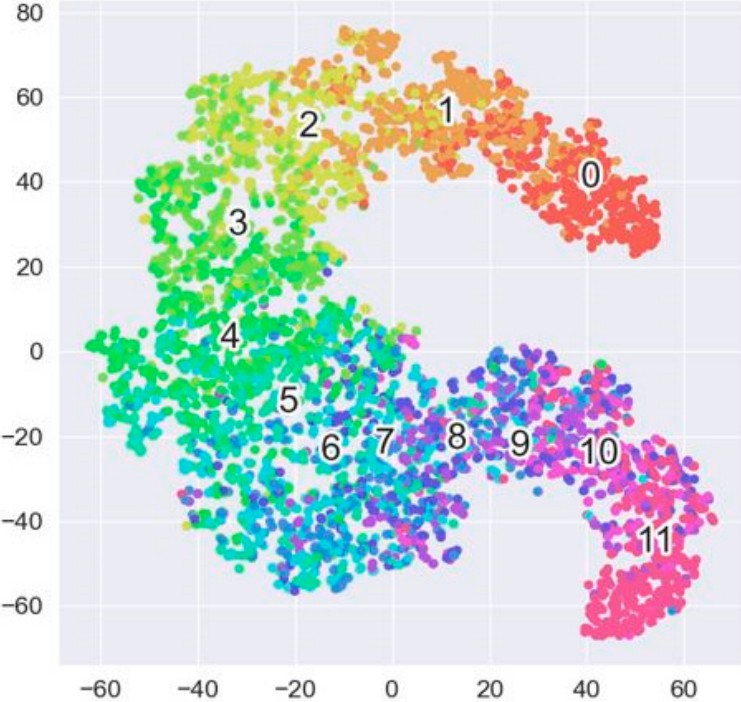

**Figure 6.** t-SNE visualization of output feature maps for 5000 training samples of the classification model. The numbers in the graphs represent the categories of number of days to ripeness of Hass avocado fruit.

### 3.3. Ablation Study

The SSRN outperformed the DenseNet overall by providing higher accuracy for the classification model and lower RMSE for the regression model (Tables 6 and 7). These features can be attributed to the SSRN being designed specifically to extract both spectral and spatial features from hyperspectral images independently and, hence, learning more discriminative features at a deeper level. We also observed that the DenseNet performed very poorly when 100% of the spectral data was fed into the model, as compared with the dimensionally reduced data (PCA) (Table 6). This model was unable to extract features, possibly because of the extremely high dimensionality of the data input to the DenseNet architecture. Hence, the model was unable to learn, leading to poor performance.

**Table 6.** Effect of PCA (amount of original spectral data used) on RMSE of SSRN and DenseNet regression models, over 200 epochs, for predicting the duration to ripeness of Hass avocado fruit.

|  | PCA (88.73% Spectral Data) | 100% Spectral Data |
| --- | --- | --- |
| SSRN RSME (days) | 2.64 | 1.32 |
| DenseNet RMSE (days) | 2.52 | 33.89 |

**Table 7.** Effect of PCA (amount of original spectral data used) on prediction accuracy of SSRN and DenseNet classification models, over 200 epochs, for predicting the duration to ripeness of Hass avocado fruit.

|  | PCA (88.73% Spectral Data) | 100% Spectral Data |
| --- | --- | --- |
| SSRN accuracy | 18.79% | 37.83% |
| DenseNet accuracy | 20.42% | 31.24% |

We further analyzed four main factors that control the performance of the trained SSRN for the regression model, as recommended previously [36]. These factors included the learning rate, the number of filters/kernels in the convolutional layers, the regularization technique, and the spatial size of the input cubes. We set the batch size to 32 and adopted the SGD optimizer that updated the model parameters iteratively by moving them in the direction of the gradient calculated on a batch of training data [49]. In the training process, only models with the best performance (lowest loss) in validation groups were saved, and the results were all generated by these optimal models.

Firstly, learning rates dictate the amount by which the weights are updated for each training iteration, i.e., the learning step. An improper learning rate could lead to divergence or very slow convergence. Therefore, we used a trial-and-error method over multiple recommended learning rates to find the optimum value. Based on the outcomes of regression, the optimum learning rate was 0.01 (Table 8).

**Table 8.** Effect of learning rate on SSRN performance over 200 epochs for predicting the duration to ripeness of Hass avocado fruit.

| Learning Rate | RMSE (Days) |
| --- | --- |
| 0.05 | 1.65 |
| 0.01 | 1.32 |
| 0.005 | 1.47 |
| 0.001 | 1.84 |
| 0.0005 | 1.63 |
| 0.0001 | 1.40 |

Secondly, the number of kernels determines the representation capacity and computational consumption of the SSRN. The network had the same number of kernels in every convolutional layer of the residual blocks (Figure 3). We examined the performance with

different numbers of kernels (Table 9). The model with 32 kernels in each convolutional layer attained the lowest RMSE, i.e., best performance. These results were obtained in 200-epoch training processes for each setting.

**Table 9.** Effect of number of kernels on SSRN performance over 200 epochs for predicting the duration to ripeness of Hass avocado fruit.

| Number of Kernels | RMSE (Days) |
|:---:|:---:|
| 8 | 2.59 |
| 16 | 2.40 |
| 24 | 2.20 |
| 32 | 1.32 |
| 40 | 2.42 |
| 48 | 2.30 |

Thirdly, deep learning models tend to overfit training data. Batch normalization and a dropout layer were used as regularization strategies when the model was in training mode. We set the dropout rate to 50%, as recommended previously [36], and evaluated the performance of the models with and without each of the above two regularization techniques for 200 epochs under the same conditions. The SSRN performed best when using both regularization strategies (Table 10). The reason for Nan (not a number) values was that we were performing regression over a neural network, because of which the output values were unbounded. This made the model prone to the exploding gradient problem, which occurs when large error gradients accumulate, resulting in large updates to the network during training, making the model unstable and unable to learn [50]. Batch normalization not only regularized the model but also eliminated the exploding gradient problem by applying a penalty to the large weights.

**Table 10.** Effect of regularization strategies on SSRN performance over 200 epochs for predicting the duration to ripeness of Hass avocado fruit.

| Regularization Strategy | RMSE (Days) |
|:---:|:---:|
| None | Nan |
| Dropout | Nan |
| Batch normalization | 1.88 |
| Both | 1.32 |

Fourthly, to assess the impact of spatialized input, we analyzed the performance of the regression model with various spatial sizes of input cubes. The model performed optimally when the spatial input was $1 \times 1$ (Table 11). This performance could be attributed to the fact that the $1 \times 1$ sized input cubes were able to learn more discriminative features as compared to larger ones, given that there was a limitation in the original number of HSI avocado samples.

**Table 11.** Effect of spatial kernel size on SSRN performance over 200 epochs for predicting the duration to ripeness of Hass avocado fruit.

| Kernel Size | RMSE (Days) |
|:---:|:---:|
| $1 \times 1$ | 1.32 |
| $3 \times 3$ | 1.35 |
| $5 \times 5$ | 1.70 |
| $7 \times 7$ | 1.36 |
| $9 \times 9$ | 1.41 |
| $11 \times 11$ | 1.84 |

## 4. Discussion

The regression model with the SSRN backbone performed best amongst all experiments. This model estimated the duration to ripeness of Hass avocado fruit with an average error of only 1.17 days.

The accuracy of the classification model was low with an average accuracy of 51.4%. Previous works have reported high classification accuracies for three broad categories of unripe, ripe, and over-ripe [5,33], whereas we categorized our data within 12 categories, which may have contributed to decreased prediction accuracy of our model. We found that there was a natural ordinal relationship between the original days-to-ripen categories. Due to this relationship, we were able to convert a conventional classification problem into a regression problem using the same underlying network architecture (i.e., SSRN). In the classification task, the ordinal relationship was not taken into consideration in the categorical cross entropy (CCE) loss function. The CCE loss was calculated based on whether the actual and predicted categories aligned. Therefore, the CCE loss calculated was the same if a fruit actually needed 3 days to ripen and the predicted value was 5 days or the predicted value was 10 days. A unified palatalization for all categories was unfavorable since the model should ideally be penalized more highly if the prediction was further away from the true label. In contrast, the MSE loss in the regression model was higher for those prediction values that were further away from the true value. This contributed to the higher quality of results from the regression model.

We analyzed the visualized output feature maps of the model to better understand the comparatively inadequate performance of the classification model (Figure 6). We observed that, while there were clear clusters formed for each category, there was strong overlap with neighboring clusters that accounted for frequent misclassification. This overlap was also seen in the confusion matrix where there were multiple predictions made within 1 or 2 days to ripeness from the true duration, which drastically reduced the overall accuracy of the model (Table 5). For this reason, we resorted to the regression-based model over the classification model.

The essence of using deep learning models is to learn the representation of input data automatically without any need for feature engineering because the models can themselves extract the discriminative features given appropriate network architectural designs and training process settings [36]. To allow for automatic feature extraction without any loss of data, we experimented without using dimensionality reduction techniques. This experiment, that fed 100% of spectral data into the model, led to better performance, although it required a longer training time. This highlights that the models performed best when they automatically learned features, given the complete spectral data, without any form of dimensionality reduction (PCA).

Both the loss and evaluation metric in the regression model were based on mean squared error (Equation (2)), but there was a slight difference between the values of the loss (MSE) and square of the RMSE metric. This difference could be explained by the regularization used in the model. In training mode, regularization was used in the different layers of the network (batch normalization) to avoid the model overfitting. Hence, the penalty applied to the weights led to a higher MSE loss value. On the other hand, the model performs evaluation in testing mode. Therefore, the weights were frozen, and predictions were made directly without any penalties added. These prediction values in testing mode were closer to true values, which explained the higher value of MSE loss (training/test dataset) when compared with the corresponding MSE evaluation metric.

Our study provided an RMSE of 1.17 days for ripening predictions, with the model with 32 kernels in each convolutional layer providing the lowest RMSE. An RGB camera has successfully been used to predict fruit firmness as an indicator of fruit ripeness without predicting the exact ripening time [27]. In a previous study, RGB spectral information provided lower prediction accuracy for nut rancidity than that obtained from hyperspectral imaging [16]. The low RMSE in our study and those reported previously [30] might be due to the ability of hyperspectral imaging cameras to capture a large number of

spectral bands from avocado fruit skin. Additionally, both the learning rate and kernel size are important for decreasing computing time and increasing prediction accuracy when developing models [51]. For example, increased kernel size decreases the prediction accuracy for dry matter concentration in avocado fruit due to the inclusion of unnecessary spatial information from sub-images such as unnecessary information in the image corners and edges [52]. Therefore, in this work, the learning rate and kernel size were optimized to achieve the best prediction accuracy.

Hyperspectral imaging successfully predicted the ripening time for avocado fruit. Ripening of avocado fruit has been extended by increasing their flesh calcium concentration [53]. We have previously predicted the concentration of calcium in the flesh from skin images of Hass fruit [26]. Thus, successful prediction of ripening time from skin images in the current study could be explained partly by the ability of hyperspectral cameras to predict internal chemical concentrations and provide information on the flesh quality of avocado fruit.

## 5. Conclusions

We developed a technique that predicted the duration to ripeness of Hass avocado fruit. We performed regression using a three-dimensional, supervised, deep learning framework. This allowed for spectral–spatial representation learning of hyperspectral images of fruit. The regression model achieved an average root mean squared error (RMSE) value of 1.17 days on the test set, indicating that the number of days to ripeness of Hass avocado fruit was estimated with an average error of only 1.17 days without any form of dimensionality reduction. Predicting the duration to ripeness with an accuracy of 1.17 days would be acceptable given that the fruit ripened over 7–15 days when held at room temperature. Our results also indicate that direct estimation of duration to ripeness via regression is a better approach than classification for estimating the ripeness of Hass avocado fruit. Fruit are usually presented on retail shelves with an unknown duration from harvest, and consumers often touch and squeeze avocado fruit to estimate the ripening stage. Information on time-to-ripeness would allow consumers to select a fruit with the desired shelf life. Our research showed great potential for combining hyperspectral imaging and deep learning to estimate the ripeness of avocado fruit, thus helping with post-harvest avocado processing, retail display, and waste reduction.

**Author Contributions:** Conceptualization, S.H.B., S.J.T., K.K. and W.K.; methodology, Y.J.D., W.K. and K.K.; software, Y.J.D.; validation, Y.J.D., S.H.B., S.J.T., K.K. and W.K.; formal analysis, Y.J.D.; investigation, Y.J.D., S.H.B., S.J.T., K.K. and W.K.; resources, Y.J.D., S.H.B., S.J.T., K.K. and W.K.; data curation, Y.J.D.; writing—original draft preparation, Y.J.D.; writing—review and editing, Y.J.D., S.H.B., S.J.T., K.K. and W.K.; visualization, Y.J.D., S.H.B., S.J.T., K.K. and W.K.; supervision, S.H.B. and K.K.; project administration, Y.J.D.; funding acquisition, S.J.T. and S.H.B. All authors have read and agreed to the published version of the manuscript.

**Funding:** This research was funded by PH16001 of the Hort Frontiers Pollination Fund, part of the Hort Frontiers strategic partnership initiative developed by Hort Innovation, with co-investment from Griffith University, University of the Sunshine Coast, Plant and Food Research Ltd., and contributions from the Australian Government.

**Data Availability Statement:** Data are unavailable due to IP arrangements.

**Acknowledgments:** We thank Costa Avocado for access to their orchards.

**Conflicts of Interest:** The authors declare no conflict of interest.

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
