# Peer review of "Estimating the Ripeness of Hass Avocado Fruit Using Deep Learning with Hyperspectral Imaging"

_horticulturae, doi:10.3390/horticulturae9050599_

Round 1

Reviewer 1 Report

Developing none-destructive technique to predict the duration to ripeness of fruit is important for fruit industry, the authors described the method of combining hyperspectral imaging and deep learning to estimate the ripeness of avocado fruit .The paper is well written and the results are clearly presented, however it could be improved in the following:

1.      Please clearly state differences between the current work and the published reports, such as, Predicting the ripening time of ‘Hass’ and ‘Shepard’ avocado fruit by hyperspectral imaging.

2.      If the same samples in this paper were used in your previous publication,”Rapid determination of nutrient concentrations in Hass avocado fruit by vis/NIR hyperspectral imaging of flesh or skin”, it would be better to connect the nutrient concentrations and ripeness together, at least for discussion.

3.      More discussions about “Based on the outcomes of regression, the optimum learning rate was 0.01” and “The model with 32 kernels in each convolutional layer attained the lowest RMSE”.

4.      Table 5. “”The numbers in green indicate the number of correct predictions for that category while the remainder are false positives or true negatives”. It seems that there is none numbers in green.

Author Response

Please also see attached file for the format of Table 5

Reviewer 1

Comments and Suggestions for Authors

Developing none-destructive technique to predict the duration to ripeness of fruit is important for fruit industry, the authors described the method of combining hyperspectral imaging and deep learning to estimate the ripeness of avocado fruit .The paper is well written and the results are clearly presented, however it could be improved in the following:

Dear Reviewer 1

The authors acknowledge the constructive comments received from the anonymous Reviewer 1. We took into account all of the comments and revised the manuscript to address the concerns and suggestions. We consider that the manuscript has been improved significantly.

We have highlighted the changes in the text. Detailed responses to all of Reviewer 1’s specific comments have been provided in this document.

Kind regards,

Associate Professor Shahla Hosseini-Bai on behalf of all co-authors.

  1. Please clearly state differences between the current work and the published reports, such as, “Predicting the ripening time of ‘Hass’ and ‘Shepard’ avocado fruit by hyperspectral imaging”.

Response

There are fundamental differences between the current study and our recent publication (Han et al. 2023). In our recent publication, we imaged 316 Hass avocado fruit only once after harvest and used those images to predict how many days it took for individual fruit to become ripe. In contrast, the current study imaged fruit every day after harvest to estimate the ripeness stage and ripening speed of fruit. The Han et al. (2023) study can be applied in processing sheds where fruit are categorized on farm based on expected ripening date whereas the current study can be applied in retail stores where the exact ripening stage and remaining ripening time need to be estimated. 

We have also clarified these differences in our Introduction. Please see:

Lines 78-93 of the revised MS: Avocado fruit are harvested when the fruit are mature but ripening occurs after harvest when fruit are taken out of cool storage and placed on shelves after harvest. It is important to predict the duration to ripeness from mature avocado fruit or from fruit at subsequent ripening stages. The ripening stage of Hass avocado fruit has been predicted non-invasively through smartphone images and hyperspectral images [27-30]. However, these approaches either predict the ripeness indirectly by estimating the firmness of the fruit [31, 32], or by classifying the fruit into a very limited number of ripeness categories of unripe, ripe, and overripe [5, 29, 33]. Avocado fruit ripening is highly variable and ripening time may vary between 6 and 15 days when fruit are placed on shelves after harvest. Therefore, classifying a fruit into an unripe category would not suggest how many days that the fruit needs to ripen. In our recent study, we were able to predict the ripening time of Hass and Shepard avocado fruit when images were captured only once from mature fruit after harvest [30]. Predicting the ripening time of mature fruit after harvest allows fruit classification on farm before sending the fruit to retail stores. However, it is also important to be able to predict the ripening time at retail stores when the fruit are placed on display with an unknown duration from harvest.

Differences between our study and Varga et al 2021 and Cho et al. 2020

Varga, L. A., Makowski, J., & Zell, A. (2021, July). Measuring the ripeness
of fruit with hyperspectral imaging and deep learning. In 2021 International
Joint Conference on Neural Networks (IJCNN) (pp. 1-8). IEEE: Cited as reference number 29 in our original MS:

Varga et al. (2021) only classified fruit into a very limited number of ripeness categories of ‘unripe’, ‘ripe’ and ‘overripe’. Avocado fruit may ripen between 6 and 15 days when they are stored at room temperature. If a fruit is classified as unripe, it does not suggest how many days that the fruit needs to ripen. We have added a citation to Varga et al. 2021 into line 84 and have amended that section to further clarify this point.

Lines 78-93 of the revised MS: Avocado fruit are harvested when the fruit are mature but ripening occurs after harvest when fruit are taken out of cool storage and placed on shelves after harvest. It is important to predict the duration to ripeness from mature avocado fruit or from fruit at subsequent ripening stages. The ripening stage of Hass avocado fruit has been predicted non-invasively through smartphone images and hyperspectral images [27-30]. However, these approaches either predict the ripeness indirectly by estimating the firmness of the fruit [31, 32], or by classifying the fruit into a very limited number of ripeness categories of unripe, ripe, and overripe [5, 29, 33]. Avocado fruit ripening is highly variable and ripening time may vary between 6 and 15 days when fruit are placed on shelves after harvest. Therefore, classifying a fruit into an unripe category would not suggest how many days that the fruit needs to ripen. In our recent study, we were able to predict the ripening time of Hass and Shepard avocado fruit when images were captured only once from mature fruit after harvest [30]. Predicting the ripening time of mature fruit after harvest allows fruit classification on farm before sending the fruit to retail stores. However, it is also important to be able to predict the ripening time at retail stores when the fruit are placed on display with an unknown duration from harvest.

Cho, B. H., Koyama, K., Olivares Díaz, E., & Koseki, S. (2020). Determination of “Hass” avocado ripeness during storage based on smartphone image and machine learning model. Food and Bioprocess Technology, 13, 1579-1587. This was cited as refence number 27 in our original MS. Please see:

Lines 78-79 of the revised MS: The ripening stage of Hass avocado fruit has been predicted non-invasively through smartphone images and hyperspectral images [27-30].

Cho et al. (2020) used RGB to predict fruit firmness as an indicator of fruit ripeness which does not indicate the time to ripeness. The fruit firmness was also assessed over 7 days. This is a different camera from hyperspectral imaging cameras and usually provides a low prediction accuracy. This has now been further clarified:

Lines 138-140 of the revised MS: Some fruit became ripe after 6 days at 21 °C and so the number of images captured per day decreased from this day onwards (Figure 2).

Lines 472-485 of the revised MS: Our study provided an RMSE of 1.17 days for ripening predictions, with the model with 32 kernels in each convolutional layer providing the lowest RMSE. An RGB camera has successfully been used to predict fruit firmness as an indicator of fruit ripeness without predicting the exact ripening time [27]. In a study, RGB spectral information provided a lower prediction accuracy for nut rancidity than that obtained from hyperspectral imaging [16]. The low RMSE in our study and those reported previously [30] might be due to the ability of hyperspectral imaging cameras to capture a large number of spectral bands from avocado fruit skin. Additionally, both the learning rate and kernel size are important for decreasing computing time and increasing prediction accuracy when developing models [51]. For example, increased kernel size decreases the prediction accuracy for dry matter concentration in avocado fruit due to the inclusion of unnecessary spatial information from sub-images such as unnecessary information in the image corners and edges [52]. Therefore, in this work, the learning rate and kernel size were optimised to achieve the best prediction accuracy.

  1. If the same samples in this paper were used in your previous publication,”Rapid determination of nutrient concentrations in Hass avocado fruit by vis/NIR hyperspectral imaging of flesh or skin”, it would be better to connect the nutrient concentrations and ripeness together, at least for discussion.

Response

The same fruit samples were not used in these two studies.  We have included further discussion to relate internal nutrient concentrations to our study. Please see:

Lines 486-492 of the revised MS:  Hyperspectral imaging successfully predicted the ripening time of avocado fruit. Ripening of avocado fruit has been extended by increasing their flesh calcium concentration [53]. We have previously predicted the concentration of calcium in the flesh from skin images of Hass fruit [26]. Thus, successful prediction of ripening time from skin images in the current study could be explained partly by the ability of hyperspectral cameras to predict internal chemical concentrations and provide information on the flesh quality of avocado fruit. 

  1. More discussions about “Based on the outcomes of regression, the optimum learning rate was 0.01” and “The model with 32 kernels in each convolutional layer attained the lowest RMSE”.

Response

A new section has been added to the Discussion to further discuss the prediction accuracy of the models. Please see:

Lines 472-485 of the revised MS: Our study provided an RMSE of 1.17 days for ripening predictions, with the model with 32 kernels in each convolutional layer providing the lowest RMSE. An RGB camera has successfully been used to predict fruit firmness as an indicator of fruit ripeness without predicting the exact ripening time [27]. In a study, RGB spectral information provided a lower prediction accuracy for nut rancidity than that obtained from hyperspectral imaging [16]. The low RMSE in our study and those reported previously [30] might be due to the ability of hyperspectral imaging cameras to capture a large number of spectral bands from avocado fruit skin. Additionally, both the learning rate and kernel size are important for decreasing computing time and increasing prediction accuracy when developing models [51]. For example, increased kernel size decreases the prediction accuracy for dry matter concentration in avocado fruit due to the inclusion of unnecessary spatial information from sub-images such as unnecessary information in the image corners and edges [52]. Therefore, in this work, the learning rate and kernel size were optimised to achieve the best prediction accuracy.

  1. Table 5. “”The numbers in green indicate the number of correct predictions for that category while the remainder are false positives or true negatives”. It seems that there is none numbers in green.

Response

Thank you for pointing out this issue. The number of correct predictions has been changed to green colour. Please see below:  

Table 5. Confusion matrix of duration of ripeness for samples of the test dataset predicted using the classification model. The numbers in green indicate the number of correct predictions for that category while the remainder are false results.

Duration to ripeness (d)

0

1

2

3

4

5

6

7

8

9

10

11

0

729

67

0

0

9

0

0

0

0

0

0

0

1

143

594

0

1

112

2

0

0

0

0

0

0

2

0

0

54

217

0

1

0

4

24

8

38

38

3

0

0

10

750

0

0

0

0

27

6

40

19

4

3

95

0

0

600

143

20

3

0

0

0

0

5

0

8

0

2

149

527

190

22

1

0

0

1

6

0

1

0

2

4

216

415

191

53

4

1

0

7

0

1

0

5

1

43

200

475

140

10

8

5

8

0

0

10

20

1

10

85

286

291

33

11

9

9

0

0

7

26

0

2

24

151

247

55

56

44

10

0

0

27

93

1

0

9

54

202

31

74

61

11

0

0

29

122

0

0

1

10

131

35

54

98

Reviewer 2 Report

This MS shows a successful application of DL in some downstream tasks, and the authors have considered comprehensively including dataset preparation, network architecture, training, and ablation experiment. Hence I recommend publishing the MS after some minor issues are addressed:

1. Symbol notations should be normalized to meet the requirement of the research paper.

2. The quality of the figures should be improved to make them clearer.

Author Response

Please also see attached file 

Reviewer 3 Report

The cultivation of avocados, due to the high water needs and the resulting burden on the environment, has been under the watchful eye of ecologists for some time. Therefore, any research aimed at increasing the efficiency of cultivation of these fruits should be known as expedient. The research presented in this article is one of them. In my opinion, this article would certainly gain in quality with the following considerations:

- the content in lines 100-106, which is a representation of the "table of contents" and the content of the article, is unnecessary;

- in the description of the location of the experiment, sometimes when it is extensive, reference is made to descriptions contained in other publications. In this paper (line 112), however, I consider it unnecessary. The description should be supplemented with a few additional sentences, and not force the reader to find it in another publication;

- the authors write (line 114) about the harvest of 80 fruits from 40 trees, how were these trees and the fruits on them selected (representativeness)? Were 2 fruits harvested from each tree, or was it done differently?

- the fruits (80 pcs.) ripened for about 11 days (line 121), each of them took one photo every day during this time (line 130), so why the number of photos was only 551 (lines 131, 141,142)?

- on what basis were the photos divided into "training, validation, testing" (lines 145-146)?

- description of the theoretical basis (in terms of the purpose of research and results) of generating subimages (lines 161-162) needs to be supplemented;

- research is assumed to be of application nature, however, the authors did not include indications for their practical application in the conclusions

Author Response

Reviewer 3

Comments and Suggestions for Authors

The cultivation of avocados, due to the high water needs and the resulting burden on the environment, has been under the watchful eye of ecologists for some time. Therefore, any research aimed at increasing the efficiency of cultivation of these fruits should be known as expedient. The research presented in this article is one of them. In my opinion, this article would certainly gain in quality with the following considerations:

Dear Reviewer 3

The authors acknowledge the constructive comments received from the anonymous Reviewer 3. We took into account all of the comments and revised the manuscript to address the concerns and suggestions. We consider that the manuscript has been improved significantly.

We have highlighted the changes in the text. Detailed responses to all of Reviewer 3’s specific comments have been provided in this document.

Kind regards,

Associate Professor Shahla Hosseini-Bai on behalf of all co-authors.

- the content in lines 100-106, which is a representation of the "table of contents" and the content of the article, is unnecessary;

Response

We have now deleted the following section from the Introduction of the MS:

‘The remainder of the paper is structured as follows. Section 2 (Materials and Methods) describes the data collection and preparation, and the deep learning approach, to estimate the ripeness of avocado fruit. Section 3 (Results) presents the experimental evaluation of the method including a comparison between the regression and classification results and an ablation study of the deep learning network. Section 4 (Discussion) provides a discussion of the results and the performance of the method. The paper concludes with a summary of findings in Section 5 (Conclusions).’ 

- in the description of the location of the experiment, sometimes when it is extensive, reference is made to descriptions contained in other publications. In this paper (line 112), however, I consider it unnecessary. The description should be supplemented with a few additional sentences, and not force the reader to find it in another publication;

Response

We have removed the citation and have now included the information on annual rainfall and maximum and minimum temperatures:

Lines 113-116 of the revised MS: The sites receive average precipitation of 1004 mm annually. The average maximum daily temperatures varied between 22.3 °C and 31.4 °C, and the average minimum daily temperatures ranged between 11.0 °C and 22.4 °C, in 2018 (Bureau of Meteorology 2023).

- the authors write (line 114) about the harvest of 80 fruits from 40 trees, how were these trees and the fruits on them selected (representativeness)? Were 2 fruits harvested from each tree, or was it done differently?

Response

We have now further clarified the method for fruit selection: 

Lines 118-120 of the revised MS: Ten mature Hass avocado fruit were harvested from each of eight trees, providing 80 fruit in total. Each tree was divided into five sectors, with one fruit harvested from the inside and one fruit harvested from the outside of the canopy in each sector.

- the fruits (80 pcs.) ripened for about 11 days (line 121), each of them took one photo every day during this time (line 130), so why the number of photos was only 551 (lines 131, 141,142)?

Response

Ripening of some fruit occurred from 6 days onwards. We did not further image a fruit once it became ripe. Hence, the number of images per day decreased after 6 days at 21 °C , as presented in Figure 2. We have further clarified this point. Please see:

Lines 138-140 of the revised MS: Some fruit became ripe after 6 days at 21 °C and so the number of images captured per day decreased from this day onwards (Figure 2).

- on what basis were the photos divided into "training, validation, testing" (lines 145-146)?

Response

The image selection was random and so ‘randomly’ has now been included. Please see:

Lines 154-155 of the revised MS: The acquired hyperspectral images were partitioned randomly into three sets: training, validation and testing.

- description of the theoretical basis (in terms of the purpose of research and results) of generating subimages (lines 161-162) needs to be supplemented;

Response

We have now further explained the basis of the experiments in the Introduction and Discussion sections. Please see:

Lines 78-93 of the revised MS:  Avocado fruit are harvested when the fruit are mature but ripening occurs after harvest when fruit are taken out of cool storage and placed on shelves after harvest. It is important to predict the duration to ripeness from mature avocado fruit or from fruit at subsequent ripening stages. The ripening stage of Hass avocado fruit has been predicted non-invasively through smartphone images and hyperspectral images [27-30]. However, these approaches either predict the ripeness indirectly by estimating the firmness of the fruit [31, 32], or by classifying the fruit into a very limited number of ripeness categories of unripe, ripe, and overripe [5, 29, 33]. Avocado fruit ripening is highly variable and ripening time may vary between 6 and 15 days when fruit are placed on shelves after harvest. Therefore, classifying a fruit into an unripe category would not suggest how many days that the fruit needs to ripen. In our recent study, we were able to predict the ripening time of Hass and Shepard avocado fruit when images were captured only once from mature fruit after harvest [30]. Predicting the ripening time of mature fruit after harvest allows fruit classification on farm before sending the fruit to retail stores. However, it is also important to be able to predict the ripening time at retail stores when the fruit are placed on display with an unknown duration from harvest.

Lines 479-485 of the revised MS: Additionally, both the learning rate and kernel size are important for decreasing computing time and increasing prediction accuracy when developing models [51]. For example, increased kernel size decreases the prediction accuracy for dry matter concentration in avocado fruit due to the inclusion of unnecessary spatial information from sub-images such as unnecessary information in the image corners and edges [52]. Therefore, in this work, the learning rate and kernel size were optimised to achieve the best prediction accuracy.

- research is assumed to be of application nature, however, the authors did not include indications for their practical application in the conclusions

Response

Thank you for the suggestion. We have now revised the Conclusion to further emphasize the implications of the study.

We have deleted the following section from the Conclusion:  

‘A limitation of our research is that the deep learning model was trained on a fixed set of training samples and, when new samples become available, the model has to be retrained from scratch. In practice, a deep learning model with the ability to learn incrementally from training data as they become available is desirable. Despite this limitation’

and replaced it with the following section: 

Lines 504-507 of the revised MS: Fruit are usually presented on retail shelves with an unknown duration from harvest, and consumers often touch and squeeze avocado fruit to estimate the ripening stage. Information on time-to-ripeness would allow consumers to select a fruit with the desired shelf life.

We have also modified the following sentence.

Lines 507-509 of the revised MS: Our research showed great potential for combining hyperspectral imaging and deep learning to estimate the ripeness of avocado fruit, thus helping with post-harvest avocado processing, retail display, and waste reduction.
